# Calibrations of Low-Cost Air Pollution Monitoring Sensors for CO, NO_2_, O_3_, and SO_2_

**DOI:** 10.3390/s21010256

**Published:** 2021-01-02

**Authors:** Pengfei Han, Han Mei, Di Liu, Ning Zeng, Xiao Tang, Yinghong Wang, Yuepeng Pan

**Affiliations:** 1State Key Laboratory of Numerical Modeling for Atmospheric Sciences and Geophysical Fluid Dynamics, Institute of Atmospheric Physics, Chinese Academy of Sciences, Beijing 100029, China; pfhan@mail.iap.ac.cn (P.H.); meihan@mail.iap.ac.cn (H.M.); 2State Key Laboratory of Atmospheric Boundary Layer Physics and Atmospheric Chemistry, Institute of Atmospheric Physics, Chinese Academy of Sciences, Beijing 100029, China; tangxiao@mail.iap.ac.cn (X.T.); panyuepeng@mail.iap.ac.cn (Y.P.); 3Department of Atmospheric and Oceanic Science, and Earth System Science Interdisciplinary Center, University of Maryland, College Park, MD 20742, USA; zeng@umd.edu

**Keywords:** low-cost gas sensors, electrochemical air quality sensors, field evaluation, single and multiple linear regression, random forest, LSTMs, environmental factors

## Abstract

Pollutant gases, such as CO, NO_2_, O_3_, and SO_2_ affect human health, and low-cost sensors are an important complement to regulatory-grade instruments in pollutant monitoring. Previous studies focused on one or several species, while comprehensive assessments of multiple sensors remain limited. We conducted a 12-month field evaluation of four Alphasense sensors in Beijing and used single linear regression (SLR), multiple linear regression (MLR), random forest regressor (RFR), and neural network (long short-term memory (LSTM)) methods to calibrate and validate the measurements with nearby reference measurements from national monitoring stations. For performances, CO > O_3_ > NO_2_ > SO_2_ for the coefficient of determination (R^2^) and root mean square error (RMSE). The MLR did not increase the R^2^ after considering the temperature and relative humidity influences compared with the SLR (with R^2^ remaining at approximately 0.6 for O_3_ and 0.4 for NO_2_). However, the RFR and LSTM models significantly increased the O_3_, NO_2_, and SO_2_ performances, with the R^2^ increasing from 0.3–0.5 to >0.7 for O_3_ and NO_2_, and the RMSE decreasing from 20.4 to 13.2 ppb for NO_2_. For the SLR, there were relatively larger biases, while the LSTMs maintained a close mean relative bias of approximately zero (e.g., <5% for O_3_ and NO_2_), indicating that these sensors combined with the LSTMs are suitable for hot spot detection. We highlight that the performance of LSTM is better than that of random forest and linear methods. This study assessed four electrochemical air quality sensors and different calibration models, and the methodology and results can benefit assessments of other low-cost sensors.

## 1. Introduction

Due to rapid economic development over the past three decades, fossil fuel consumption has substantially increased, and thus, the associated air pollutants, e.g., carbon monoxide (CO), nitrogen dioxide (NO_2_), sulfur dioxide (SO_2_) [1,2], particulate matter (e.g., PM_2.5_ and PM_10_) [3,4], and greenhouse gases (e.g., carbon dioxide (CO_2_)), have also increased in China [5,6]. According to the World Health Organization (WHO), pollutants, including particulate matter, nitrogen dioxide, sulfur dioxide, and ozone, pose the highest risk to human health [7]. Exposure to air pollution, such as ground-level ozone, even in the short term, can affect the respiratory system [8]. Many related studies have linked air pollution with serious health problems, poor birth outcomes, and even premature death [9,10,11]. Pollutants have received public attention in China since the 2000s, and central and local governments have begun to implement stricter laws on pollutant reductions and monitoring since that time, including “China’s Action Plan of Prevention and Control of Air Pollution” in 2013 [12,13]. For example, with the implementation of the action plan, the emissions of SO_2_, nitrogen oxides (NO_X_), and PM_2.5_ decreased in 2017 by 36%, 31%, and 30% from the 2012 levels in Beijing–Tianjin–Hebei, respectively [12]. These achievements cannot be realized without accurate and widespread monitoring of air pollutants at environmental observatories. At a standard station of the Ministry of Ecology and Environment (MEE), reference instruments based on differential optical absorption spectrometry (DOAS) theory are used [14,15]. The working principle of DOAS involves using the narrow-band absorption spectrum of molecules to distinguish gas compositions and deduce the concentrations of measured gases based on the intensity of its absorption spectrum. The theoretical basis of DOAS is the Lambert–Beer law. Moreover, these research grade instruments generally require a cabin with an air conditioner to maintain the temperature (Temp) at approximately 25 °C, and high precision and accuracy are ensured by using people trained in regular instrument checks and data quality control [16].

However, due to the large spatial-temporal heterogeneity of air pollutants in urban areas resulting from variations in emission sources and atmospheric transport [17,18], it is difficult to monitor pollutants at finer resolutions, such as at the street scale, which is an urgent need for both MEE grid level reductions for fine management and the public need for local pollution information. With the development of low-cost electrochemical air quality sensors, this monitoring has become possible in recent years, especially for low- and middle-income countries [19]. Low-cost sensors integrated in Internet of Things (IoT) applications are operated in dense sensor networks for urban air quality monitoring of pollutant gases, particulate matter, and greenhouse gases with high spatial and temporal resolutions [20,21]. Sensor networks and mobile monitoring on ground vehicles and unmanned aerial vehicles [22,23] or tethered balloons [24] are presented for smart and sustainable cities and as supplements to expensive air quality monitoring stations.

Electrochemical sensors are based on a chemical reaction between gases in the air and the electrode in a liquid inside a sensor [25]. Szulczynski et al. (2017) [26] reviewed the detection of odorants (with thresholds ranging from <1 ppb for sulfur compounds to dozens of ppm for ethanol) in the vicinity of municipal processing plants using electronic and bioelectronic sensors. Low-cost sensor networks have become an important complement to regulatory-grade instruments. The advantages are outlined as follows: (1) due to the low cost, sensors can be deployed as a dense network to capture high variations; and (2) sensors are more easily deployed than standard instruments with smaller volumes and masses. The disadvantage is that the sensor’s accuracy requires corrections in ambient environments, and sensors need to be replaced at 1–2 year intervals due to chemical material consumption by the sensor [27]. Linear (zero “check” and span “check”) and multiple linear regression (MLR) and machine learning models are the most widely used methods to calibrate low-cost sensors [28,29]. Using Alphasense as an example, several studies have evaluated and calibrated sensors for NO_2_, SO_2_, or O_3_ using simple or multiple linear regression or neural network methods. Using support vector regression and random forest methods, the root mean square error (RMSE) of the corrected results for NO_2_ with a reference instrument was <5 ppb in ambient environments, and the R^2^ was between 0.74 and 0.95; these methods can be used to detect 8–10 ppb differences in NO_2_ with 90% confidence [30]. Spinelle et al. (2014) [31] used MLR and artificial neural network (ANN) methods to calibrate multiple gas sensors and found that single linear regression (SLR) and MLR produced the highest measurement uncertainty. By colocation with regulatory-grade instruments on the island of Hawaii, Hagan et al. (2019) [32] used a nonparametric algorithm (k nearest neighbors) to correct the raw signal for SO_2_, and the RMSE was <7 ppb. Previous studies generally focused on one or a few species, and some studies lacked adequate calibration.

The objectives of this study were to (1) measure and assess multiple pollutant gases (CO, NO_2_, O_3_, and SO_2_) in heavily polluted areas and (2) calibrate and validate their performances in a field environment using SLR, MLR, random forest regressor (RFR), and long short-term memory (LSTM).

## 2. Materials and Methods

### 2.1. Sensor Configuration and Field Deployment

We focused on the performance and calibration of the Alphasense gas sensor through a field comparison experiment between these four types of sensors and the national control monitoring station of the MEE. As shown in Figure 1, two models were deployed in October 2019 at the Institute of Atmospheric Physics, Chinese Academy of Sciences (IAP, CAS) (39°47ʹ N, 116°57ʹ E) tower station, which is located in Jiandemen, Haidian District, Beijing, within 2 km of the Olympic Stadium Center MEE site (39°98′ N, 116°40′ E). The monitoring site is located in the urban residential area of Beijing, surrounded by roads, so it is considered to be a typical traffic site.

We used a homemade multipollutant measuring instrument known as the SENSE Model S to conduct the experiments. Based on a literature review, we selected Alphasense sensors (B4 for CO, NO_2_, OX, and SO_2_) as the research objects, which are suitable for outdoor environments, while the A4 series is suitable for where space is critical, such as spaces with mobile air quality monitors [33,34,35,36]. Alphasense sensors measure gas pollutants (CO, NO_2_, O_3_, and SO_2_), and the sensor produces electronic current (in nA) and is converted to amplified voltage by a circuit board. We used both the original Individual Sensor Board (ISB) from Alphasense [37] and a domestic substitute named MMS as a potentiostat to transmit data (MMS is only half the price of the ISB). This potentiostat provides a dual channel voltage output from both the working electrodes (Vw) and auxiliary electrodes (Va), and these two voltages are collected by a Python script and stored in the microprocessor. Vw and Va were converted into air quality units (ppb) according to the formula provided by the manufacturer when collecting sensor data. Both the working electrode and auxiliary electrode have equivalent two-stage amplifiers, and there are no adjustments on the ISB for zero offset and gain. However, after a careful analysis of the correlation between Va and temperature, Va is positively related to the temperature change rate and not the temperature itself, and the performance is much better without Va (Appendix A). Therefore, the manufacturer’s correction for Va is not effective and thus was removed from our corrections. The raw data of sensors mentioned below refers to Vw. We designed two versions of motherboards for imported and domestic sensor boards, which also hold other sensors. The motherboard is connected to a microprocessor named BeagleBone Green Wireless (BBGW) through general-purpose input/output (GPIO) pins [38], and meteorological data, including temperature and relative humidity (RH), were collected from the Adafruit BME280 sensor. All data were collected with a time resolution of 2 s, then stored in the microprocessor and later uploaded to a remote server through the IoT, and data were also stored in a built-in secure digital memory (SD) card daily for backup. The date and time were automatically synchronized from the Alibaba Cloud through an ntpd service and ensured by a real-time clock (RTC) module in case of poor service.

### 2.2. Data Processing

Since the B4-OX sensor measured signals of both O_3_ and NO_2_, the O_3_ concentration was calculated by subtracting the B4-NO2 concentration from the B4-OX concentration. The B4 sensor had a warm-up time of approximately 2 h [39]. Therefore, data collected during the first 2 h after powering on were excluded from this study. The raw second data were first quality control checked by eliminating outliers that were beyond 3 sigma and then averaged into minute data and hourly data. We collected the MEE data measured at official monitoring stations from the China National Environmental Monitoring Center website (http://www.cnemc.cn/) [40]. The data obtained from MEE were at 1-h resolution. The raw gas sensor signal outputs were collected at 2-s resolution and resampled to 1-h resolution using Python function pandas. Data Frame. Resample () for comparison with reference measurements. The statistical calculations and figure plots were performed using the Python Scipy package.

### 2.3. Evaluation Parameters

We used three parameters to evaluate the performance of the sensors. The coefficient of determination (R^2^) (Equation (1)) is calculated by the square of R. It is used to reflect the relationship between the low-cost sensor (sensor) and the reference instrument (reference). The value of R^2^ ranges from 0–1, and the larger the R^2^ value is, the better the correlation relationship between the sensor and the reference is. The RMSE is used to measure the deviation between the observed value and the true value. In this study, the deviation between the sensor and the reference was evaluated to reflect whether the sensor is reliable under severe field conditions. The percentage relative bias (Equation (2)) was also calculated to determine the measurement error of the sensor compared with the reference.
(1)$$R^{2}={\left[\frac{\sum^{\;}\left(sensor-\bar{sensor}\right)\left(reference-\bar{reference}\right)}{\sqrt{\sum^{\;}{\left(sensor-\bar{sensor}\right)}^{2}\sum^{\;}{\left(reference-\bar{reference}\right)}^{2}}}\right]}^{2}$$
(2)$$Relative\;bias=\left(\frac{sensor-reference}{average\;\left(reference\right)}\right)*100\%$$

### 2.4. Calibration Methods

#### 2.4.1. Simple Linear Regression and Multiple Linear Regressions

Two linear calibration models were built: simple linear regression (SLR) and multiple linear regression (MLR). The calibration by an SLR used Equation (3). C_raw is the value measured by the low-cost sensor. C_correction is a model-predicted (calibrated) value. SLR was performed for different sensors, and the a, b coefficients applicable to each sensor were obtained. The MLR used C_raw, temperature, and RH measured by the low-cost sensor as predictors because the low-cost sensor only measured these parameters. The MLR model is expressed as Equations (4)–(6). To include as many environmental factor ranges as possible, we used one set of sensors as the training data and the other set of sensors as the test.


SLR: C_correction = a*C_raw + b
(3)


MLR: C_correction = a*C_raw + b*Temp + c
(4)


C_correction = a*C_raw + b*RH + c
(5)


C_correction = a*C_raw + b*Temp + c*RH + d
(6)

#### 2.4.2. Random Forest Regressor

The random forest regressor (RFR) is a very mature and widely used regressor [41]. It works by constructing a multitude of regression trees during training time. RFR has been used in sensor data correction in many studies and has achieved good results [42,43].

We implemented six selected variables, including working electrodes (Vw) and auxiliary electrodes (Va), concentration data calculated by the original factory formula (Alpha), temperature (Temp), and RH data of the sensor to train a random forest machine learning regression model. To prevent overfitting, we limited the maximum depth of each tree. For CO data, the maximum depth of trees was set to 8, the number of trees was set to 300, and the criterion function was the mean absolute error. For O_3_ and NO_2_ data, the maximum depth of trees was set to 6 and 9, respectively, the number of trees was set to 300, and the criterion function was also the mean squared error. Similar to the MLR method, we used one set of sensors as the training data and the other set of sensors as the testing data in the RFR model.

#### 2.4.3. Long Short-Term Memory Networks

Among all deep learning applications, recurrent neural networks (RNNs), as well as long short-term memory (LSTM) neural networks, are commonly utilized to handle various problems with temporal information. In each LSTM cell, one hidden layer represents the output information at the previous time and provides short-term memory similar to RNN cells, and another cell state carries long-term memory. The forget gate helps to discard some information that is not essential. Therefore, LSTM can process longer time series information better than RNN. At present, some studies have applied LSTMs to data processing, correction, and prediction, and good results have been obtained [44,45,46,47].

In the LSTM model used in this study, we used seven selected variables (Time, MEE, Alpha, Vw, Va, Temp, RH) as inputs. The LSTM model used in our study had one LSTM layer and one fully connected layer. The variables of the input layer corresponded to Alpha, Vw, Va, Temp, and RH with timesteps of 25, 3, and 29 (which means we used the data of the first 24/2/28 h to predict the data of the 25/3/29th h) for NO_2_, O_3_, and CO data, respectively. The size of the mini batch for each epoch was set to 200. The detailed configuration of the model is expressed in Table 1. The MEE data were used as the target to supervise the neural network process. Similar to the MLR and RFR methods, we used sensor 1 as the training data and sensor 2 as testing data in the LSTM method.

## 3. Results and Discussion

### 3.1. Performances of Sensors with Linear Regressions (SLR/MLR)

The two sets of instruments were deployed in the field for approximately one year. Figure 2 shows the raw data and SLR correction results of the two sets of sensors just after deployment for half a year before April 2020. It can be seen from the raw data that the two sensors, especially O_3_ and SO_2_, have different degrees of zero drift, but the trends of the two sensors are very consistent. CO, O_3_, and NO_2_ are very close to MEE after SLR correction. Because of the low SO_2_ concentrations in the environment, the signal monitored through SLR correction cannot achieve the same effect as the other three. Appendix A is the complete uncalibrated original time series of the two sets of instruments. The data quality before April 2020 was good, but after April 2020, the performance of the sensors deteriorated due to the temperature rise and the potential impacts of sensor life. This result can be confirmed in Table 2, which shows the changes in the stability and biases of the four gas sensors in one of two models after deployment by calculating the seasonal statistics of the sensor including R2, RMSE, and slope and intercept when applied with SLR.

From the statistical results, the performance of these four sensors varied with the seasons, and R^2^ had different degrees of attenuation. Among them, CO decreased from R^2^ = 0.85 in the autumn of 2019 to R^2^ = 0.64 in the summer of 2020. Then, the temperature dropped in the autumn of 2020, and the impact of the high temperature on the sensors was weakened, so R^2^ increased to 0.67. Similarly, the R^2^ of O_3_ decreased from 0.74 in the fall of 2019 to 0.35 in the summer of 2020. In the fall of 2020, no statistical results were obtained due to missing MEE O_3_ data. The NO_2_ result went from R^2^ = 0.76 to R^2^ = 0.12, and in the fall of 2020, it rose slightly to R^2^ = 0.32, which was similar to the CO result.

The SO_2_ sensor performance was the worst among these sensors, with R^2^ values ranging from 0.00 to 0.01 in Table 2. This result is not surprising since Beijing had very low SO_2_ concentrations (with an annual mean value of 4 ppb for MEE and 75% of the MEE data were generally <4 ppb). According to the datasheet of Alphasense SO2-B4 [36], the noise (or resolution) is 5 ppb, and performances are tested from 0–200 ppb for general use. For MEE, one type of reference instrument generally used is Thermo Scientific Model 43 i, with a lower detectable limit <0.5 ppb and noise (in RMSE) of 0.25 ppb with a 300 s averaging time [48]. These values indicated that Alphasense is not suitable for such low-concentration monitoring, while the MEE reference instrument is qualified. The low SO_2_ concentrations are largely attributed to the strong implementation of the SO_2_ control policy; thus, good monitoring and treatment measures were achieved at the stacks near large individual emissions factories [49], and coal consumption was largely reduced due to the shift to natural gas [50]; these measures reduced the SO_2_ emissions by 62% from 2010–2017 [51]. Due to the low SO_2_ environmental concentrations and subsequent poor performance of the sensor response, we mainly analyze CO, O_3_, and NO_2_ in the following text.

In addition to the R^2^ values in Table 2, the slope and intercept of the sensors and MEE also varied greatly with season, which reflects the high sensitivity and poor stability of the sensor to the environment. Many studies have also mentioned that environmental monitoring instruments, especially low-cost electrochemical instruments, are prone to drifting and performance degradation as the environment and service life change. The CO, O_3_, and NO_2_ sensor data can be corrected by SLR within half a year to achieve ideal results (e.g., R^2^ > 0.75); however, it is often improbable to maintain this effect for a long time (e.g., >6 months). For electrochemical sensors, the correction effect was different with time due to the impact of seasonal environmental changes and the consumption of chemical materials by the sensor; therefore, time-varying coefficients are recommended at seasonal time intervals.

We also conducted MLR training and tests. The MLR method included the influences of environmental factors, i.e., temperature (Temp) and relative humidity (RH). Table 3 shows the performances of the CO sensor with linear regressions (SLR and MLR), and Appendix A show those of O_3_ and NO_2_, respectively.

Generally, we found that there were no obvious improvements in terms of R^2^ and RMSE after including these environmental factors in the MLR model, and R^2^ was even slightly decreased after including both Temp and RH compared with SLR. For the CO training data, the R^2^ was maintained at 0.83 before and after MLR Temp or RH correction, but a worse performance (R^2^ = 0.79) was seen when including Temp and RH. Although there were no changes to R^2^, the SLR and MLR methods substantially improved the performance of O_3_ because the RMSE decreased from 734 ppb to 217–295 ppb after corrections. Using sensor 2 as the test set, R^2^ performed better than sensor 1 because of its shorter deployment time in the field and because it was less affected by high temperatures in summer. Using the model trained by sensor 1 to test sensor 2, the results obtained were basically the same as the results of the training set. After the SLR and MLR corrections, R^2^ did not change, and after adding Temp and RH simultaneously, R^2^ decreased from 0.85 to 0.81. The O_3_ and NO_2_ results were basically the same as the CO results.

Therefore, MLR generally did not improve the performances of these sensors after considering the influences of temperature and RH compared with SLR.

### 3.2. Calibration by Machine Learning (RFR)

We used the machine learning method (RFR) after failing in the use of linear methods such as SLR and MLR to eliminate temperature and humidity noise. Figure 3 shows the results of the CO, O_3_, and NO_2_ data calibrated by RFR. For CO, as shown in Figure 3a, this method increased R^2^ from 0.83 to 0.92 and from 0.85 to 0.89 for the training procedure and testing procedure, respectively. This result shows that the improvement in CO by the RFR method is not obvious since the SLR already performed well. When the training model is applied to the test set, the test set does not increase significantly, which shows that the characteristics of different CO sensors are different, and the applicability of the RFR model is limited. For O_3_ sensor calibration in our study (Figure 3b), the corrected results of the training set and test set are R^2^ = 0.74 (R^2^ = 0.54 before correction) and R^2^ = 0.64 (R^2^ = 0.59 before correction). However, the high value of O_3_ after correction will be significantly underestimated, especially in the test set, which is a shortcoming of the model from Figure 3b. For NO_2_ (Figure 3c), the R^2^ of the training and testing data increased from 0.35 and 0.44 to 0.85 and 0.75, respectively, which proved that the RFR model significantly improved the NO_2_ sensor.

RFR has been used in sensor data correction in many studies and has achieved good results. In Zimmerman’s research [42], the RF model was applied to low-cost sensors and was proven to accurately characterize air pollution concentrations at the low levels typical of many urban areas in the United States and Europe. The R^2^ of the models in his research was 0.91 for CO, approximately 0.86 for O_3_, and approximately 0.67 for NO_2_, which was close to our research results. Bigi [30] also applied three algorithms, including multivariate linear regression, support vector regression, and random forest, and demonstrated that RF is the best correction algorithm, with R^2^ reaching 0.79 for NO_2_ correction.

### 3.3. Calibration by Neural Network (LSTMs)

After only using the information of one certain time to perform calibration, we propose a deep learning approach that contains temporal characteristics for our regression task. In this part of our work, we show how LSTM neural networks can be employed to correct the raw sensor data. Figure 4 shows the results of the CO, O_3_, and NO_2_ data calibrated by LSTM networks.

For CO (Figure 4a), LSTM had further improvements, mainly reflected in the testing procedure (R^2^ = 0.93), compared with RFR (R^2^ = 0.89), as shown in Figure 4a. This result shows that the improvement in CO by the LSTM method is more obvious, and when the training model is applied to the test set, the test set is also improved, which shows that the applicability of the LSTM model to different CO sensors is better than RFR.

In the research of Spinelle [48], an Alphasense-O_3_ B4 sensor was used, and its effect after MLR correction reached approximately R^2^ = 0.5. There are few existing studies on correcting the low-cost sensor of O_3_ using deep learning and other algorithms, and the control of O_3_ pollution has attracted increasing attention from the public and scientific community [49], so the correction and application of a low-cost sensor for O_3_ is urgently needed. For O_3_ sensor calibration in our study (Figure 4b), the corrected results of the training set and test set are R^2^ = 0.75 (R^2^ = 0.54 before correction) and R^2^ = 0.77 (R^2^ = 0.59 before correction), and the LSTM correction effect of O_3_ is significantly improved compared with RFR, especially in the test set. However, from Figure 4b, the high value of O_3_ after correction is also significantly underestimated, especially in the test set, which is also a shortcoming of the LSTM model.

For NO_2_ (Figure 4c), the R^2^ of the training and testing data increased from 0.35 and 0.44 to 0.85 and 0.84, respectively, which proved that the LSTM model significantly improved the NO_2_ sensor compared with RFR. Using nonlinear algorithms (support vector regression and random forest), Bigi et al. (2018) [30] found a better performance of the Aphasense NO2-B4 sensor for urban sites, with an RMSE < 5 ppb, an R^2^ between 0.74 and 0.95 and an mean absolute error (MAE) between 2 and 4 ppb. Bigi et al. also found that all algorithms exhibited a drift (mean daily residual) that ranged between 5 and 10 ppb for random forest and 15 ppb for MLR at the end of deployment. NO_2_ concentration differences of 8–10 ppb were reliably detected, depending on the level of air pollution.

A separate analysis of SO_2_ was conducted. The concentration signal measured by the sensor was very weak because the environmental concentration of SO_2_ was very low (~40 ppb) and greatly affected by the environment. The use of SLR and MLR to correct SO_2_ had no effect (R^2^ is basically 0). Therefore, we selected a period of time with the best quality from the SO_2_ observation data of one of the sensors, with 50% for the training set and 50% for the testing set. Appendix A shows the results of the SO_2_ data calibrated by LSTMs. The R^2^ of the sensor and MEE before training was 0.15, and the R^2^ was 0.36 after training. For the test set, R^2^ increased from 0.17 to 0.33. Therefore, the LSTM model improved the SO_2_ sensor more significantly than the linear model, but it still did not reach the accuracies of the CO, O_3_, and NO_2_ sensors. In the research of Hagan [52], in an environment near a volcano, where the SO_2_ concentration is as high as 1000 ppb, the correction effect of the SO_2_ sensor after applying the linear and K-Nearest Neighbor (KNN) algorithms reached R^2^ = 0.99. Therefore, the sensor performs poorly at low concentrations, but it is worth continuing to explore monitoring and early warnings of high concentrations.

In general, for CO, O_3_, NO_2_, and SO_2_, the best agreement between sensors and reference measurements was observed for neural network calibrations compared with the linear and multilinear regressions and RFR. This shows that these sensor data have some nonlinear characteristics that we have not yet understood. The neural network method can actively learn data features, so using AI methods can quickly solve some problems in the case of not fully clarifying the data features.

### 3.4. Sensor Biases under Different Pollution Conditions

To further compare the differences between the traditional linear method and AI method, we evaluated the sensor performances under different pollution levels and different temperature and RH conditions using SLR corrected data and LSTM corrected data.

The responses of electrochemical air quality sensors under different pollution conditions are very important for providing complimentary spatial information to standard instruments, especially under high pollution conditions. Most of the previous studies were conducted in countries and regions with good air quality [28], such as the United States and Europe. This study illustrated the performance of low-cost CO, NO_2_, O_3_, and SO_2_ sensors at high concentrations. In reference to the MEE [53] value for different pollution conditions (Appendix A), we evaluated the sensor performances under different pollution levels. For SLR, there were either positive or negative relative biases for the three sensors, while LSTMs successfully overcame this disadvantage and maintained a close mean bias of approximately zero for all three types of sensors. Moreover, the SLR method had obvious positive biases under low concentrations, which is not a large problem since the value under clean air can tolerate relatively large errors and would not surpass the first-grade critical value. CO showed the best performances for all these sensor types, with mean relative biases decreasing from 10% to −5% under concentrations of 0–500 to >3000 ppb for SLR (Figure 5a). CO is not a major polluting species in Beijing and is part of the “Excellent” level (Appendix A) most of the time. Moreover, CO measurement is also useful as a tracer for fossil fuel combustion and other pollutants due to their homology and needs further investigation in future studies. For O_3_, the relative biases decreased from 90% to −10% under concentrations of 0–40 to >150 ppb for SLR (Figure 5b). For O_3_ light pollution and moderate pollution (Appendix A), the SLR was not good enough, while the LSTMs decreased the biases to <5%. The mean relative biases for NO_2_ decreased from 180% to −20% under concentrations of 0–15 to >100 ppb for SLR (Figure 5c). The NO_2_ pollution in Beijing is also light and is primarily less than 150 ppb (excellent to good).

### 3.5. Impacts of Different Environmental Factors on the Calibration Results

Temperature influenced electrochemical air quality sensors in a complicated manner by affecting the working electrode and auxiliary electrode [52,54], as well as the resistance and capacitance sensors on ISBs or motherboards; thus, there was no universal law for the responses of the three sensors to temperature in this study. To compare the two calibration methods, there were relatively large positive or negative biases for the three sensors for SLR, while the LSTMs performed much better. For CO and NO_2_, there were increasing trends of relative biases with the increase in temperature and an abrupt drop when the temperature surpassed 40 °C (Figure 6a,c), which is due to the sharp decline in sensor measurements when the temperature is >40 °C (Appendix A). Specifically, for CO, the relative biases were −10%, 0%, 22%, 24%, and −10% under temperatures of <10 °C, 10–20 °C, 20–30 °C, 30–40 °C, and >40 °C for SLR, respectively (Figure 6a). The mean relative biases for NO_2_ were −2%, 1%, 40%, 40%, and −120% under temperatures of <10 °C, 10–20 °C, 20–30 °C, 30–40 °C, and >40 °C for SLR, respectively (Figure 6c). For O_3_, the relative biases were 25%, 2%, −20%, −10%, and 40% under temperatures of <10 °C, 10–20 °C, 20–30 °C, 30–40 °C and >40 °C for SLR, respectively (Figure 6b). For the LSTMs, the relative biases were less than 10% in most cases (Figure 6a–c).

Humidity affects the reaction activities of different gas species [54]. CO is an inert gas, and thus, its relative biases showed no trend with increasing RH for either the SLR or LSTM models. O_3_ showed a decreasing trend of relative biases, and NO_2_ showed an increasing trend for the SLR method. In detail, the relative biases were less than 5% for all the RH conditions (Figure 7a). For O_3_, the relative biases decreased from 25% to 0% for RHs of <20% and 20–40%, while they increased to −25% for RH conditions >40% using the SLR method (Figure 7b). For NO_2_, the relative biases decreased from 40% to approximately 0% from RH < 20% to RH = 60% and then increased to 75% for RH > 75% with the SLR (Figure 7c). For the LSTM method, the relative biases were less than 5% for O_3_ and less than −50% for NO_2_. Hagan et al. [52] also found that RH was negligible for Alphasense sensors, although the sensor response may be dependent on RH; for certain specific datasets, RH does not contribute to the signal uniquely due to the strong coverage of temperature. In the experiments of Bigi et al. [30], RH was proven useful in NO regressions.

The comparisons between the simple linear correction and the neural network show that the neural network works well under various environmental conditions and is a promising method for sensor calibration when SLR cannot achieve satisfactory performances.

## 4. Conclusions

In this paper, four types of electrochemical air quality sensors, namely, CO, NO_2_, O_3_, and SO_2_ were comprehensively studied in a field environment in Beijing. We used linear regression methods (SLR and MLR), machine learning methods (RFR), and neural networks (LSTMs) to train and test the measurements with nearby reference measurements from MEE. We found that the MLR generally did not improve the performances of these sensors after considering the influences of temperature and RH compared with SLR. However, the RFR and LSTM models significantly increased the performances of O_3_, NO_2_, and SO_2_ compared with the SLR and MLR, while CO did not increase significantly since the SLR already performed well. The SO_2_ sensor did not perform as satisfactorily as the other three sensors, largely due to the low environmental gas concentrations. However, in general, for CO, O_3_, and NO_2_, the best agreement between sensors and reference measurements was observed for neural network calibrations (LSTM) compared in all methods.

In addition, we evaluated the sensor performances under different pollution levels. For the SLR calibrated data, there were either positive or negative relative biases for the three sensors, while the LSTMs successfully overcame this disadvantage and maintained a close mean bias of approximately zero for all three types of sensors (CO, NO_2_, and O_3_). Moreover, the SLR method had obvious positive biases under low concentrations, which is not a large problem since the value under clean air can tolerate relatively large errors and would not surpass the first-grade critical value. We found that systematic biases of the CO, O_3_, and NO_2_ sensors are small for the SLR and LSTM calibrated data at high concentrations, indicating that the sensors have good responses and high measurement accuracies under high concentration conditions. The results also show that after calibration, these sensors are suitable for hot spots with high pollution.

Regarding the impacts of different environmental factors on the calibration results, we found that there was no universal law for the CO, O_3_, and NO_2_ sensor responses to temperature and RH in this study. The three sensors had relatively large positive or negative biases for the SLR calibrated data, while the LSTMs performed much better, which indicated that the LSTMs obviously eliminated the influence of temperature and humidity on the sensor signals. For CO and NO_2_, there were increasing trends of relative biases with increasing temperature and an abrupt drop when the temperature surpassed 40 °C.

The methodology and results have extensive potential benefits and implications for assessing other low-cost sensors. We have evaluated the performances of the sensors over the course of one year, and much longer study periods are still needed to evaluate the length of a chemical sensor’s life. Three of the four sensors we evaluated (CO, O_3_, and NO_2_) are suitable for pollution hotspot monitoring with LSTM calibrations.

## Figures and Tables

**Figure 1 sensors-21-00256-f001:**
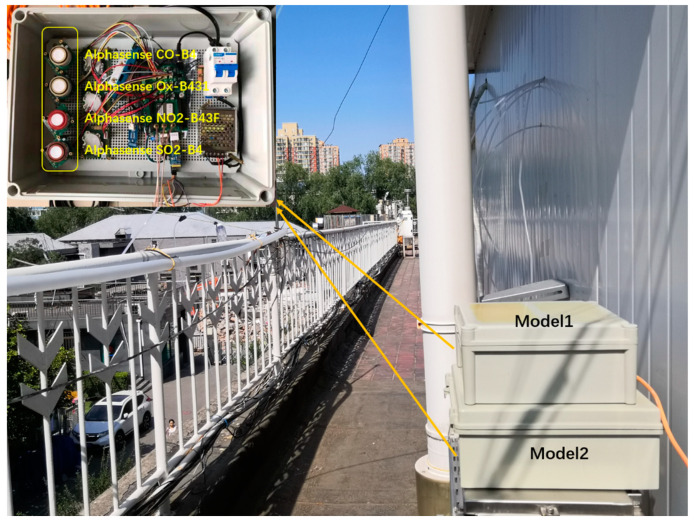
Photographs of multipollutant measuring instruments Model 1 and Model 2.

**Figure 2 sensors-21-00256-f002:**
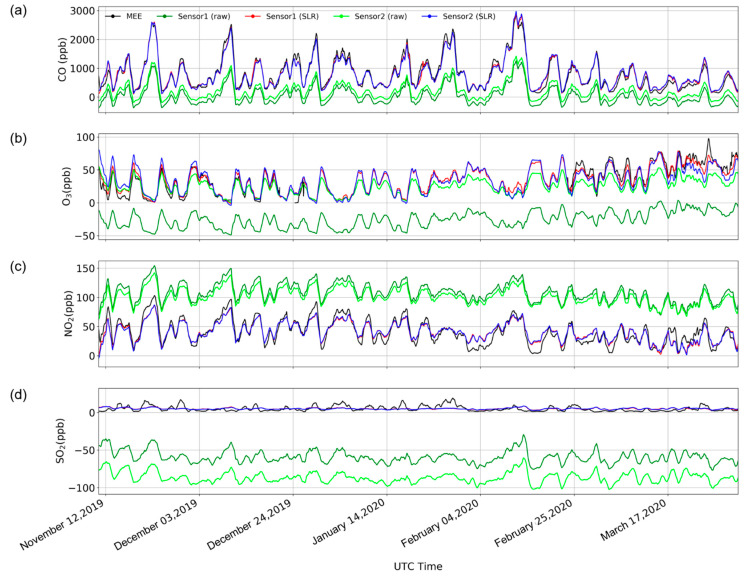
Comparison of (**a**) CO, (**b**) O_3_, (**c**) NO_2_, and (**d**) SO_2_ volume concentrations between the Ministry of Ecology and Environment (MEE) Olympic Sports Center station and the two sensors before (raw) and after single linear regression (SLR) calibration on 10 November 2019, to 1 April 2020, in Beijing based on a 24 h rolling average.

**Figure 3 sensors-21-00256-f003:**
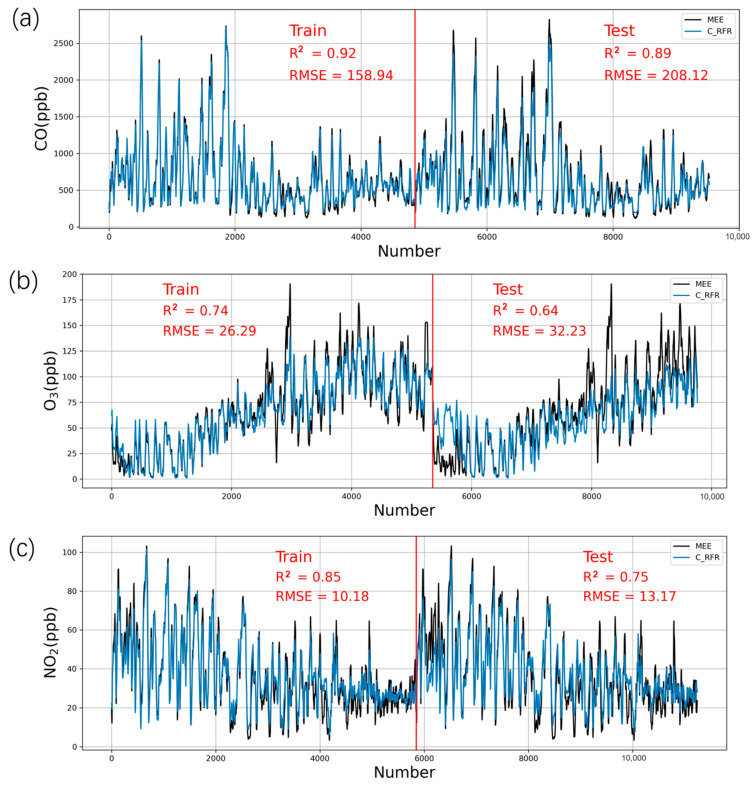
Training and testing of sensors using the random forest regressor (RFR) method and comparison with MEE data from the reference instruments based on a 24 h rolling average. (**a**) CO, (**b**) O_3_, and (**c**) NO_2_. One set of sensors in Model 1 is the training data, and the other set of sensors in Model 2 is the testing data.

**Figure 4 sensors-21-00256-f004:**
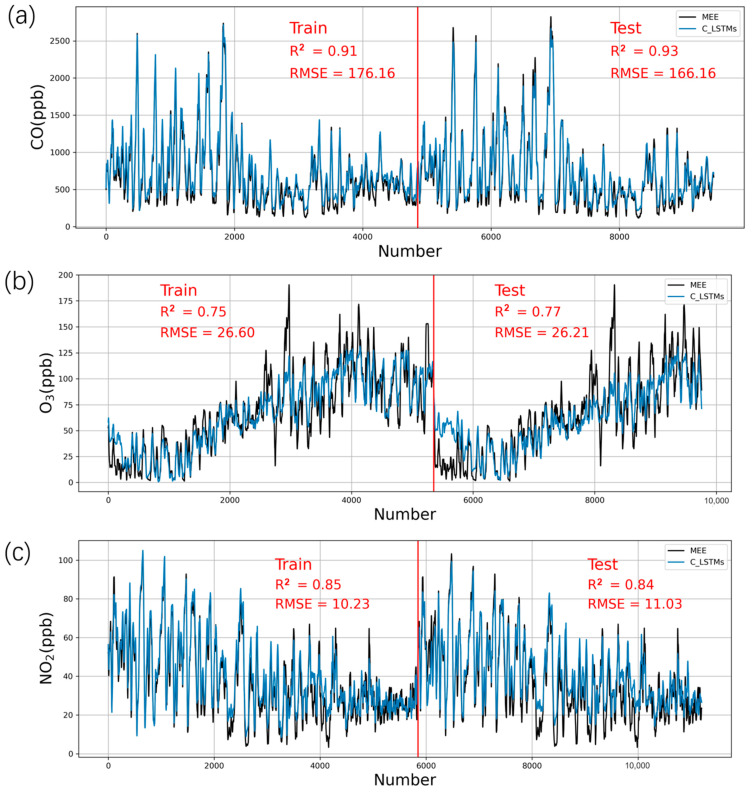
Training and testing of sensors using the long short-term memory (LSTM) method and comparison with MEE data from the reference instruments based on a 24 h rolling average. (**a**) CO, (**b**) O_3_, and (**c**) NO_2_. One set of sensors in Model 1 is the training data, and the other set of sensors in Model 2 is the testing data.

**Figure 5 sensors-21-00256-f005:**
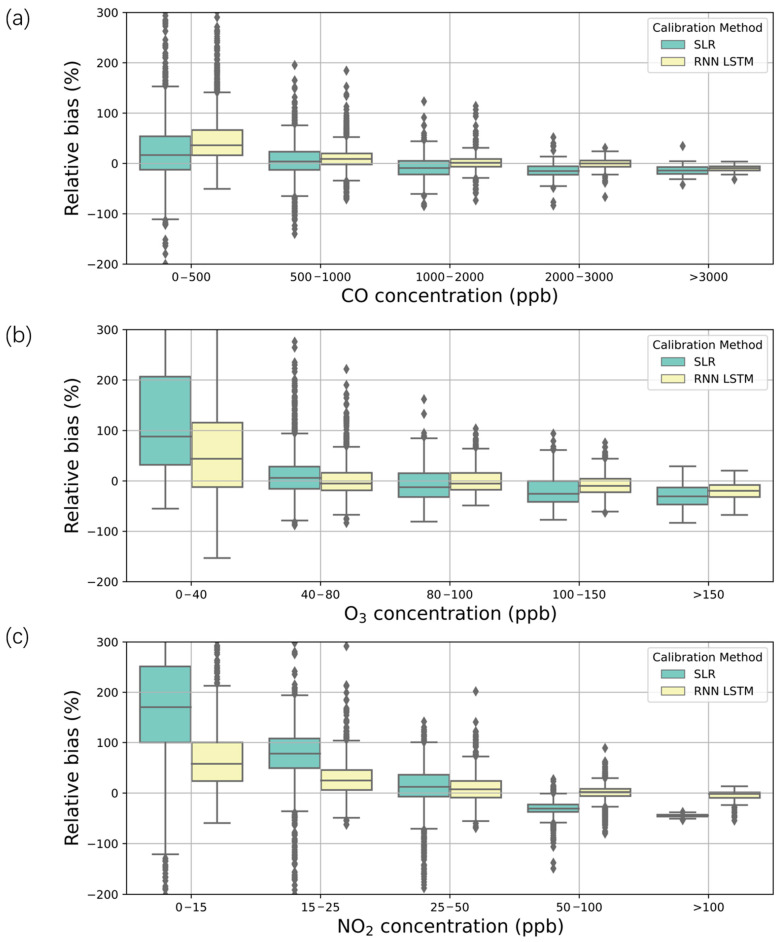
Comparison of the relative biases between SLR/LSTM calibrated sensors and MEE at different concentration ranges. Boxes of different colors represent different correction methods. The lines within the boxes represent the medians of data. The interquartile range (IQR) is the range of the boxes and indicates the difference from the 25th to the 75th percentile. Whiskers represent values within 1.5 times the IQR, and dots are values outside 1.5 times the IQR. (**a**) CO, (**b**) O_3_, and (**c**) NO_2_.

**Figure 6 sensors-21-00256-f006:**
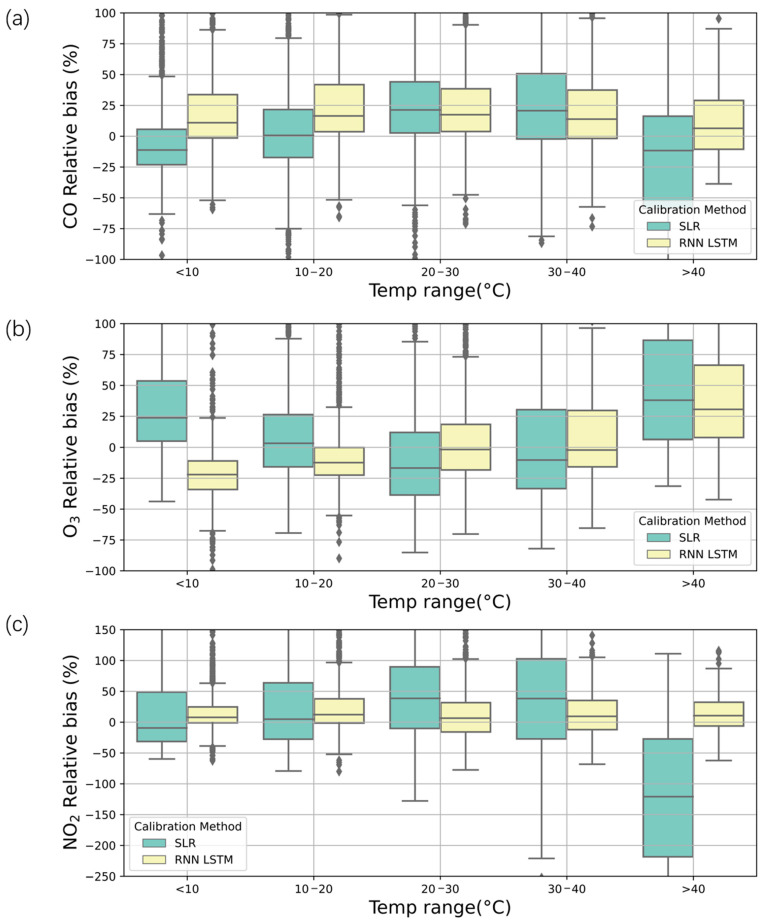
Comparison of the relative biases between the calibrated sensors and MEE at different temperature ranges. (**a**) CO, (**b**) O_3_, and (**c**) NO_2_.

**Figure 7 sensors-21-00256-f007:**
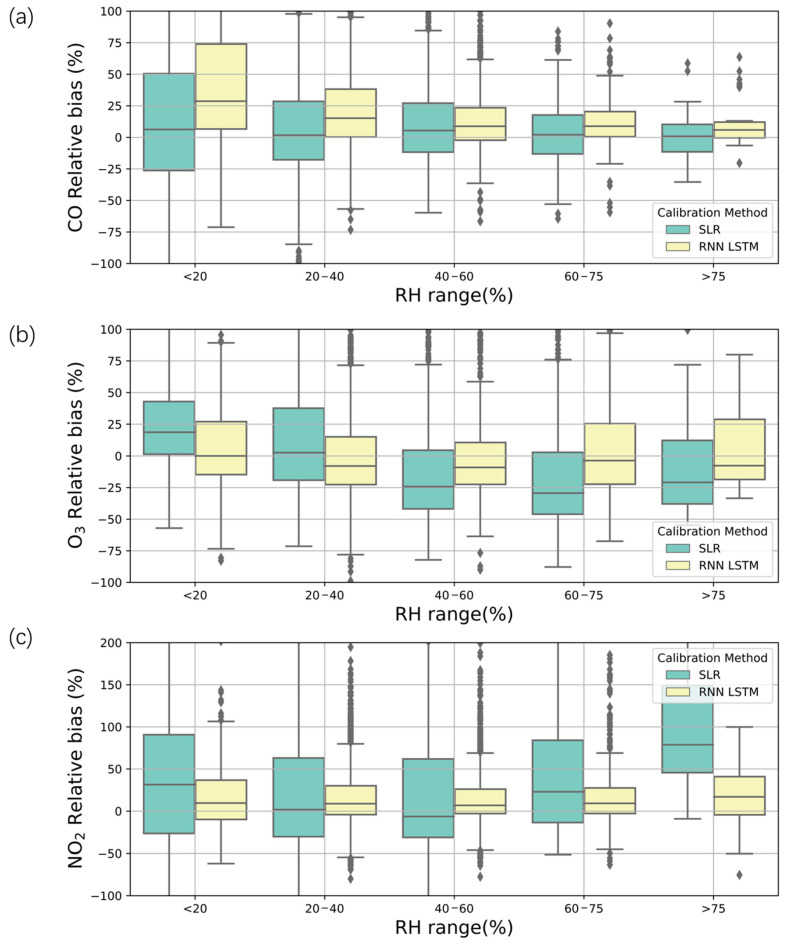
Comparison of the relative biases between the calibrated sensors and MEE at different relative humidity (RH) ranges. (**a**) CO, (**b**) O_3_, and (**c**) NO_2_.

**Table 1 sensors-21-00256-t001:** Detailed configuration of the neural network.

Configuration	Value
Number of hidden layers	10
Number of neurons in the hidden layer	150
Input variable	Time, MEE, Alpha, Vw, Va, Temp, RH
Number of the output variable	1
Training data percentage	50%
Validation data percentage	50%
Data normalization	Minmax
Training algorithm	Long short-term memory networks

**Table 2 sensors-21-00256-t002:** Seasonal changes in the stability and biases of four sensors in one model after deployment.

Pollutant	Statistic	Autumn (October–November 2019)	Winter (December 2019–February 2020)	Spring (March–May 2020)	Summer (June–August 2020)	Autumn (September–October 2020)
CO	number	633	1513	1559	957	665
	R^2^	0.85	0.88	0.79	0.64	0.67
	RMSE	232	239	156	168	145
	slope	1.41	1.71	1.32	1.03	1.11
	intercept	701.73	767.28	598.10	579.68	567.64
O_3_	number	884	627	2140	1705	0
	R^2^	0.74	0.79	0.45	0.35	--
	RMSE	11	12	34	43	--
	slope	1.34	1.59	1.31	0.89	--
	intercept	60.45	74.93	80.35	95.97	--
NO_2_	number	866	2096	1890	1118	754
	R^2^	0.76	0.77	0.28	0.12	0.32
	RMSE	13	12	18	17	17
	slope	0.93	1.22	0.32	0.12	0.32
	intercept	−51.2	−94.05	−2.11	15.82	1.33
SO_2_	number	866	2096	1890	1118	752
	R^2^	0.01	0.05	0.00	0.00	0.00
	RMSE	5	5	3	1	1
	slope	−0.03	0.11	0	0	−0.01
	intercept	4.04	12.44	3.77	2.64	2.2

**Table 3 sensors-21-00256-t003:** Performances of the CO sensor with linear regressions (SLR and MLR).

Linear Regression Calibration Model	Train Data		Test Data	
	before Correction	after Correction	before Correction	after Correction
C_correction = a*C_raw + b		R^2^ = 0.83		R^2^ = 0.85
		RMSE = 227.85		RMSE = 242.33
C_correction = a*C_raw + b*Temp + c		R^2^ = 0.83		R^2^ = 0.86
	R^2^ = 0.83	RMSE = 217.19	R^2^ = 0.85	RMSE = 221.72
C_correction = a*C_raw + b*RH + c	RMSE = 734.13	R^2^ = 0. 83	RMSE = 788.68	R^2^ = 0.85
		RMSE = 227.85		RMSE = 242.33
C_correction = a*C_raw + b*Temp + c*RH + d		R^2^ = 0.79		R^2^ = 0.81
		RMSE = 295.10		RMSE = 325.25

## Data Availability

The data presented in this study are available on request from the corresponding authors.

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
