# Peer review of "Calibrations of Low-Cost Air Pollution Monitoring Sensors for CO, NO2, O3, and SO2"

_sensors, 2021, doi:10.3390/s21010256_

Round 1

Reviewer 1 Report

This manuscripts presents a year-long study of low-cost air quality sensors co-located with reference-grade instruments in China. There is a growing literature on this topic, but there are still many unknowns and this study is particularly valuable since it occurred in an area with relatively high pollution. A highlight of the analysis is comparing conventional linear regression techniques with more modern data analysis approaches, including a neural network.

I have a relatively large number of minor comments below. There is a lot of good material in the current manuscript, but it needs substantial changes and revisions before publication. In addition to these minor comments, more detail about the how the actual measure occurs is necessary. What is actually measured? Current? Voltage, as implied by Figure S1? Is how is this converted to air quality units, as seen in Figure 2? Is there a faculty default calibration? Or, has the voltage simply been scaled? Next, some of the language is a bit colloquial. I have noted this specifically in several instances, but more generally, the manuscript should be carefully edited for scientific writing.

Minor comments

Lines 32-35: I assume this sentence refers specifically to China, which is mentioned explicitly in the second sentence. Make clear this statement is about China, explicitly, in the first sentence.

Line 46: Maybe “grade” instead of “gate?”

Line 88: How is air pushed or pulled through the instrument? Is there an active fan?

Lines 91-97: Several points here. First, more detail is necessary. What is the function of the function of these boards besides “collecting the raw signals?” Need to know how the signal is collected. For the domestically-sourced ISB, should provide a block diagram or a circuit diagram of how the raw signal is converted to a digital measurement. Second, the data presented in Figure S1 and the associated analysis should be in Results, not Materials and Methods. It is confusing to jump back and forth.

Line 106 and following: I assume this is all done in post-processing, not in the instrument. Please make this clear.

Lines 111-112: What do you mean by “resampled?” Give the specific details on the approach used to put both datasets into the same time series.

Lines 118-119: Is this the same as information about line 106? This should be consolidated into one place. Also, you record every 2 second, average to one minute for the 3-sigma calculation, and then average again to one hour? This should be more clear.

Line 128: This equation for r2 is a confusing, since you are using the subscript “i” for different gases and you also have summations. Since the equation is so common, it’s fine to omit it.

Lines 129-130: Why does sensor have a subscript “i” but reference doesn’t?

Line 135: Need to define exactly what the training dataset was.

Line 136-137: You used temperature & RH from the Alphasense itself or from the Adafruit board?

Line 153 (Table 1): the variables in the table seem to be different from what is defined in the text. For example, what is “Alpha?”

Line 156 (Figure 1): What exactly is C_raw? Is it really already in ppb units? It’s not a current? Or, is it the value using the default calibration from the factory (Alphasense)?

Lines 164-165: Using this time series plot, it’s difficult to see this drift. It would be much better to have another plot that shows either the absolute difference (instrument – reference) or percent difference.

Lines 170-171: This description of Table 1 is not clear, especially since Table 1 has not been introduced previously.

Lines 172-178: It is speculative to say that temperature caused this deterioration in fit. I am sure that pollutant values also changed while temperature changed. The lack of fit at higher temperatures could be due other covarying factors.

Lines 179-186: give the factory specification for the AlphaSense instrument and compare it to the measured MEE values.

Lines 192-195: these statements go too far given the observations in the current study. First, remove ‘impossible’ and give language that draws the implications from your study. Also, the reasons given in lines 193-194 may be true, especially the consumption of the sensors, but this is speculation.

Lines 200-201: Again, need more information about the training data, and this should be given in the Materials and Methods.

Lines 204-206: Isn’t the AlphaSense output already corrected for temperature? So, this is not surprising. Be more clear on this point.

Lines 207-209: If RMSE went down but the r2 didn’t change, that implies the sensor was not correctly calibrated, correct?

Lines 210-211: Clarify this result. This is good, right?

Lines 226-228: Again, this should be in Materials and Methods and more clearly explained.

Lines 238-241: this conclusion seems out of place. Also, the language could be improved. For example, “hot and difficult point” are not precise terms to use in the scientific literature.

Lines 246-247: How is this ‘drift’ extracted from the model?

Line 259: To make this statement more meaningful, what is the evidence that the sensor accurately responded to high so2 values measured by the MEE? Conversely, did the AlphaSense instrument often give ‘false positives’: that is high values that were not concurrently measured by the MEE?

Line 275: “levels” instead of “grades”

Lines 270-288: I like this analysis, but isn’t it in the nature of the neural network to remove these biases? Also, while CO may be “excellent” in terms of health impacts, the measurement is also used as a tracer for fossil fuel combustion and other pollutants.

Lines 280-281: Don’t give the numbers in the text, since they are already in the figures. This goes for all the following instances too. If there is a particularly relevant number, that’s OK to include. But, don’t give all the values that are coming straight from the figures.

Lines 306-308: This should be better connected to the earlier discussion around line 95. See my note on lines 91-97.

Lines 313-320: Again, do not just give the numbers that can be seen in the graphs. Instead, give some context and analysis around this result. Other studies have seen relationships with RH. Why is it not observed here?

Author Response

We thank the reviewer for understanding of this work and valuable comments. We believe the constructive feedback will improve the paper and increase its potential impact to the community.

Reviewer 2 Report

1.In my opinion there are moreover a few changes to be made by authors to render more clearly their work.

(1)There is an extra symbol in line 34, ")".

(2) Many references in the paper are not indexed in order, such as [13,14,25] in line 62, please check other places.

(3) The paper lacks the organizational structure of each section.

(4) Line 89 should be O3, right?

(5) The content of section 2.3.1 is not aligned left and right according to the format, right?

2.The methods used in the paper are general and lack innovation. It is recommended that authors use innovative methods and make comparisons to highlight the characteristics of the paper.

Author Response

Reviewer 2 suggested that the results can be presented in a more clear way, we have made corrections in our manuscript according to the Reviewer’s comments.

Reviewer 3 Report

Comments in the attached file

Author Response

Thanks to the reviewer for the good comments and suggestions. We have made corrections in our manuscript according to the Reviewer’s comments.

Round 2

Reviewer 1 Report

Thank you for the thoughtful responses to my comments from the first review. I am satisfied that almost all of my concerns were addressed.

Author Response

Thank you for your careful review. We have made some revisions (e.g., lines22, 40, 68, 73, etc.) for the language of the manuscript.

Reviewer 2 Report

The author did not make changes according to the first comment. The same problem is not corrected in this version.

  1. The reference index in this article does not appear in order, such as [29] of line 78, [26] of line 87, and the same problem in many other places.
  2. The element does not use subscripts, such as line 137, line 258,...
  3. The data in Table 2 is not formatted.
  4. The font of the reference is not formatted.
  5. The number of Figure S2 of Line 281 is wrong, and the corresponding figure is not found in the text.

Author Response

  1. Response to comment: The reference index in this article does not appear in order, such as [29] of line 78, [26] of line 87, and the same problem in many other places.

Thank you for your careful review. We have carefully checked the manuscript and all the references in the introduction have been adjusted in order. But in the result and discussion, some papers have been indexed in the above introduction (e.g., line81), there are still some jumps from [51] (line227) to [30] (lines283,312).

  1. Response to comment: The element does not use subscripts, such as line 137, line 258,...

Thank you for your careful review. The subscripts in line185 have been revised. Since the element (B4-OX, B4-NO2) in lines124-125 are sensor name, we does not use subscripts in reference to relevant datasheet[1, 2].

  1. Response to comment: The data in Table 2 is not formatted.

Thank you for your careful review. Revised accordingly.

  1. Response to comment: The font of the reference is not formatted.

Thank you for your careful review. Revised accordingly.

  1. Response to comment: The number of Figure S2 of Line 281 is wrong, and the corresponding figure is not found in the text.

Thank you for your careful review. Corresponding figure S2 is in the supplementary materials, so it was not concluded in the text.

Figure S2. Comparison of (a) CO, (b) O3, (c) NO2 , (d) SO2 volume concentrations between the MEE Olympic Sports Center station (MEE) and the two uncalibrated sensor, (e) and (f) are temperature and relative humidity of the field where the sensor packages is located on October 25, 2019 to October 10, 2020, in Beijing based on a 24 h rolling average.

Reference:

  1. Alphasense Alphasense NO2-B43F Nitrogen Dioxide Sensor Datasheet. Available from http://www.alphasense.com/WEB1213/wp-content/uploads/2019/09/NO2-B43F.pdf (Accessed on Jan 2019). 2019.
  2. Alphasense Alphasense OX-B431 Ozone + Nitrogen Dioxide Sensor Datasheet. Available from http://www.alphasense.com/WEB1213/wp-content/uploads/2019/09/OX-B431.pdf (Accessed on Jan 2019). 2019.
